# Targeting Aberrant RAS/RAF/MEK/ERK Signaling for Cancer Therapy

**DOI:** 10.3390/cells9010198

**Published:** 2020-01-13

**Authors:** Ufuk Degirmenci, Mei Wang, Jiancheng Hu

**Affiliations:** 1Division of Cellular and Molecular Research, National Cancer Centre Singapore, 11 Hospital Crescent, Singapore 169610, Singapore; 2Cancer and Stem Cell Biology Program, Duke-NUS Medical School, 8 College Road, Singapore 169857, Singapore

**Keywords:** RAS GTPases, RAF family kinases, Ras/RAF/MEK/ERK signaling, BRAF(V600E), RAF inhibitors, paradoxical activation, protein–protein interactions, synthetic lethal, neoplasm

## Abstract

The RAS/RAF/MEK/ERK (MAPK) signaling cascade is essential for cell inter- and intra-cellular communication, which regulates fundamental cell functions such as growth, survival, and differentiation. The MAPK pathway also integrates signals from complex intracellular networks in performing cellular functions. Despite the initial discovery of the core elements of the MAPK pathways nearly four decades ago, additional findings continue to make a thorough understanding of the molecular mechanisms involved in the regulation of this pathway challenging. Considerable effort has been focused on the regulation of RAF, especially after the discovery of drug resistance and paradoxical activation upon inhibitor binding to the kinase. RAF activity is regulated by phosphorylation and conformation-dependent regulation, including auto-inhibition and dimerization. In this review, we summarize the recent major findings in the study of the RAS/RAF/MEK/ERK signaling cascade, particularly with respect to the impact on clinical cancer therapy.

## 1. A Brief History of RAS/RAF/MEK/ERK Signaling Cascade

The period from 1964 to the 1980s was the era of oncogene discoveries; prominent viral oncogenes such as Ras (rat sarcoma) and Raf (rapidly accelerated fibrosarcoma) were identified during this time [1,2,3]. Shortly after their discovery, these viral oncogenes were shown to be altered versions of normal cellular genes [2,4,5,6]. Three cellular Ras genes encode the four members of the RAS family of small GTPases, KRAS4A, KRAS4B, HRAS, and NRAS, whose mutations drive one-third of human cancers. Instead of point mutations, V-RAF is an N-terminal truncated version of the cellular RAF gene (CRAF) that encodes a serine/threonine kinase [7,8]. These milestone discoveries instigated our current understanding of the dominant cancer signaling pathway: RAS/RAF/MEK/ERK. 

In 1984, it was first suggested that epidermal growth factor (EGF) stimulated activation of RAS, i.e., increased its GTP bound state [9], which linked RAS to the upstream receptor tyrosine kinase (RTK) signaling. The subsequent finding of RAS requirement for serum-stimulated DNA replication further solidified its role at the plasma membrane as a signal transducer [10,11]. In the effort to unravel the function of RAF proteins, an early study showing that v-RAF could stimulate S-phase entry in the absence of RAS activity [11] suggested that RAF functions as either downstream of RAS or in parallel to RAS to promote cell division. Studies of RAF in *Drosophila* [12] and *C. elegans* [13] confirmed its role in RTK signaling, which put RAF under RTKs and RAS. In separate studies, the cytoplasmic Ser/Thr kinases ERK1 and ERK2 were found to promote cell cycling [14,15,16,17]; and ERK1/2 activity was shown to be enhanced by yet other cytosolic kinases, MEK1/2, that phosphorylate the conserved Thr/Tyr in the activation loop of ERK1/2 [18]. Further investigation of the kinase cascade revealed that CRAF is the upstream kinase that phosphorylates MEK1 at Ser222 and MEK2 at Ser218 that regulates the activity of MEK, and through which ERK [19,20], thus rank-ordering the MAPK signaling from RAS, RAF, MEK, and finally to ERK [21]. The RAS GTPase is “switched on” to the GTP-bound active form by upstream regulators, such as RTKs, activated Ras can then physically interact with RAF and turn on the signaling cascade [22,23,24,25]. These findings delineated how signals generated from membrane-bound receptors are conveyed through RAS GTPase and transmitted intracellularly through a kinase cascade, setting a milestone in understanding of cell communication and signaling (Figure 1). 

For RAF research, the early spotlight on CRAF was shifted to BRAF after the discovery in 2002 that BRAF mutations (especially BRAFV600E) were dominant in cancer [26]. Recent studies, however, have brought CRAF back to the center stage for its role in the complicated regulation of RAF kinases by the so called inhibitor-induced paradoxical activation of RAF seen in RAF and RAS mutant cancers [27]. A main therapeutic challenge in treating RAS/RAF-driven cancers is to develop drugs that can inhibit this pathway without paradoxical activation. 

There are several major reviews in the field that describe the importance of RAS and RAF signaling and their roles in cellular regulatory processes. In this paper, we refer to these reviews, at times, due to the abundance of original research articles. However, we do provide short summaries of crucial aspects of the field, with their primary references, where we feel it enhances the clarity of this review.

## 2. RAS GTPases and Their Activation

The mammalian RAS GTPases consist of three gene isoforms, HRAS, NRAS, and KRAS, and four protein isoforms (splicing isoforms of KRAS give rise to KRAS4A and KRAS4B proteins). Whilst the isoforms share most of their sequence, substantial differences appear in the C-terminal so-called hypervariable regions and in post-translational modifications [28,29,30]. These differences in sequence and modification are considered to underlie the findings that RAS isoforms can function differentially in different biology and pathophysiology [31,32,33,34,35]. From the standpoint of engaging MAPK signaling, KRAS is more efficient than HRAS for CRAF activation while the opposite is true for PI3K activation [36]. Furthermore, both KRAS and HRAS have higher activity toward NFκB activation in contrast to NRAS [37]. While being members of the most frequently mutated oncogene family in human cancer [38], RAS isoform mutants are clearly not equally prevalent in cancers [30]. KRAS mutations are overwhelmingly represented in cancers as whole compared to the other two isoforms. There is also strong tissue predilection of the occurrence of RAS isoform mutations; while KRAS monopolizes pancreatic cancers, NRAS mutants dominate melanoma and AML. Furthermore, the RAS isoforms also have different favored mutations, which interplay with cancer types and tissue origins [38], adding complexity and intrigue [31]. All these differences among RAS isoforms underscore the limitation of our understanding of RAS proteins and their downstream pathway engagements [33]. 

The cellular activities and functions of RAS proteins are regulated at different levels. As a GTPase, RAS activity is regulated by the GTP/GDP cycle [39]. GTP-bound RAS proteins adopt the so-called active conformation that allows them to bind and activate downstream effectors, while the GDP-bound RAS proteins have altered conformations that impede such interactions. The process of dislodging bound GDP for GTP, thereby activating RAS, is facilitated by guanine exchange factor (GEF) proteins. The intrinsic GTPase activity, enhanced by RAS GTPase activating protein (GAP) [40], hydrolyses the bound GTP into GDP and returns the protein to the inactive GDP-bound state, thus completing the GTP/GDP cycle. As an important signaling process, the GTP/GDP cycle is regulated by various stimuli, including the cell surface receptors, such as several RTKs. Genetic alterations that affect the regulation of RAS activation/inactivation cycle, particularly ones that result in the persistent activation of RAS, can lead to human pathologies—the so called RASopathies [41]. For example, upregulation or gain-of-function mutations of RTKs stimulate the activity of the RasGEF, Sos [42], which in turn elevates the cellular level of GTP-bound Ras and oncogenic transformation. On the other hand, the loss-of-function mutations of RasGAPs, exemplified by NF1 [43], also results in persistent RAS activation and proliferative diseases. The most common mutations leading to RAS activation are on RAS itself, which occur in one-third of human malignancies. There are two hot spots of RAS activating mutations: the mutations at glycines 12 and 13 (G12/13) that impair RAS association with RasGAPs and at glutamine 61 (Q61) that diminish the intrinsic GTPase activity of Ras [38].

The proper functions of RAS proteins are also subject to the regulation of their posttranslational modifications, which are essential for their trafficking, membrane localization, and interaction with regulators and effectors [28,44,45,46,47,48]. RAS proteins belong to group of proteins that contain the C-terminal CAAX consensus sequence for the prenylation processing [48,49]. The nascent RAS proteins in the cytosol are firstly modified by either protein farnesyltransferase or geranylgeranyltransferase on the cysteine residue of their CAAX box [44,50,51], after which they transiently associate with the endoplasmic reticulum (ER) [51,52,53]. On the ER, they are further processed by the endoprotease RCE1 [54], followed by isoprenylcysteine carboxylmethyltransferase ICMT [55], which converts the carboxyl-terminus of RAS proteins from a hydrophilic region into a hydrophobic one, facilitating the insertion of Ras proteins into cellular membranes [56,57]. Subsequently, RAS proteins can undergo isoform-specific modifications, such as phosphorylation for KRAS4B and palmitoylation for HRAS and NRAS, which facilitate transport and plasma membrane localization through distinct mechanisms [29]. 

Interestingly, the functions of RAS are subject to another layer of regulation—dimerization and nano-clustering on plasma membrane to trigger and transmit signaling downstream [58]. Lipid rafts, subdomains of plasma membranes that have distinct chemical composition and properties, have been known since 1998 [59] and were observed as lateral heterogeneity, and consequently non-random distribution, of proteins by proteolipid-based sorting. Earlier works in Ras nano-clustering showed that Ras proteins gather on lipid rafts differentially among isoforms [60,61]. The current model is that all Ras isoforms in their GDP and GTP bound state give rise to distinct conformations and interact with distinct lipid compositions, cholesterol, PS, PA, PIP_2_, PIP_3_, PI_3_P, and PI_4_P, and that lipid composition contributes to the stability of the nanoclusters of Ras [62]. RAS dimerization or nano-clustering is thought to be a key step in the generation of its ability to couple to RAF [61,62,63]. 

## 3. RAF Isoforms

RAF proteins, pivotal components of Ras/RAF/MEK/ERK signaling cascade, include three isoforms: CRAF, BRAF, and ARAF. CRAF (also called Raf-1) is the first RAF protein identified in 1984, followed by ARAF [64,65] in 1986 and BRAF [66] in 1988. All RAF proteins share three conserved regions: CR1 (conserved region 1) [67,68], which contains a RAS-binding domain [69,70,71,72] (RBD) and a Cys-rich domain [73]; CR2, which is characterized by Ser/Thr-rich sequence; and CR3, which is constituted of a putative kinase domain with an acidic N-terminus (NTA) [74,75] and a regulatory C-terminus [76,77]. Structures of RAF kinases revealed that the proteins could be divided into two functional regions as the regulatory domain (CR1 and CR2) and kinase domain (CR3) [78,79]. Although they have similar molecular structures, RAF proteins have quite different activity and play differential roles in cell function. BRAF, which is well-known in cancer, as it is a major target of genetic mutations in tumorigenesis, has the highest activity among three isoforms, likely by virtue of its constitutively-phosphorylated NTA motif [75,80]. CRAF, which plays an indispensable role in RASopathies [81,82,83], has intermediate activity. Lastly, ARAF is rarely seen genetically altered and has the lowest kinase activity due, for the most part, to its non-canonical APE motif [84,85] (Figure 2a). 

As typical protein kinases, RAFs contain a kinase domain that comprises an N-lobe (amino-terminal lobe) and a larger C-lobe (Carboxyl-terminal lobe). The N-lobe that includes five antiparallel β-strands and a signal-integrated helix denoted as αC is connected by a flexible hinge to the C-lobe that mostly consists of α-helices and a key loop termed activation segment/activation loop. Although the kinase domains oscillate between closed and open conformations via inter-lobe motions, the active state is restricted to the closed conformation. The closed conformation is maintained by the alignment of two parallel hydrophobic cores (spines) of spatially conserved residues spanning N- and C-lobes [87,88]. The regulatory spine (R-spine) is built upon the alignment of four residues in RAFs, F516 from β4 strand, L505 from αC, F595 from DFG, and H574 from HRD (BRAF residue numbers used here) [80,89,90]. The catalytic spine (C-spine) is formed to control kinase activation upon ATP loading. These hydrophobic spines limit the movement of αC helix and stabilize the active conformation of the kinase domain [91,92]. 

In inactive conformation of RAF, the “Asp-Phe-Gly (DFG)” motif at the N terminus of the activation loop is flipped “OUT” relative to its conformation in the active state “IN”. The “OUT” DFG blocks the ATP binding pocket and stabilizes an open inactive conformation [93]. The αC helix adjacent to N-terminus of the dimer interface is also in an “OUT” position in the inactive kinase, where it binds to dimer interface residues (504–511 in BRAF) and further stabilizes the inactive conformation [87,94]. In addition, F485 from the β3 strand, L597 on the activation loop, and V600 on helix AS-H1 form a hydrophobic network that provides further stabilization of inactive conformation [95]. In the active state, αC assumes the “IN” conformation that helps the formation of a salt bridge between glutamine from αC and catalytic lysine from VAIK motif (E501-K483 for B-Raf, E393-K375 for C-Raf, and E354-K336 for A-Raf) [96]. The activation loop AS-H1 is disordered in the active conformation due to αC being pulled “IN”, which reciprocally enables dimerization [95]. It is important to note that the hallmark of an active RAF kinase requires the assembly of the two hydrophobic spines in its core, which supersedes the more dynamic/oscillating elements such as the so-called “open” or “closed” states or formation of the conserved salt bridge [88]. 

## 4. RAF-Knockout Mouse Models

Gene-knockout mouse models of the three RAF proteins have yielded a vast amount of information on their functional differences. CRAF-knockout mice are embryonic lethal with poor development of the placenta, liver, and hematopoietic organs [97,98]; BRAF-knockout mice die in utero at D12.5 with vascular and neuronal defects [99,100]. In contrast, ARAF-knockout mice were born alive, but presented severe intestinal and neurological abnormalities [101].

Interestingly, mice that express a kinase-impaired CRAF with the loss of phosphorylation sites in NTA motif (YY340/341FF) exhibit a different phenotype from that caused by complete CRAF-deficiency [102]. Several mechanisms have been suggested to explain potential kinase-independent functions of CRAF. Firstly, Rok-α is hyperactivated and mis-localized to the membrane in CRAF^−/−^ cells, which blocks Fas death receptor internalization, leading to increased Fas on the plasma membrane and Fas-dependent apoptosis [103,104]. Furthermore, CRAF NTA mutant mice show impaired wound healing and migration of keratinocytes [104]. CRAF-deficient fibroblasts and keratinocytes exhibit rounded morphology and impaired migration, indicating the defective cytoskeleton. This phenotype could be rescued with Rok-α double knockout. Biochemical and functional evidence support a mechanism for these phenotypes of kinase-independent actions of CRAF being that the autoinhibitory Cysteine-rich domain of CRAF acts in trans to inhibit Rok-α. This involvement of CRAF in the regulation of apoptosis has been shown to be relevant in several human diseases. For example, Rok-α inhibition by CRAF is a prerequisite for Ras-induced cellular transformation [81,82,105,106,107], and it has been shown that CRAF plays a role in pathogen-mediated macrophage apoptosis and erythroid differentiation [108,109].

ERK signaling can inhibit apoptosis in various ways, including expression of caspase inhibitors, neutralization of Bcl-2 family proteins [110,111], and activation of NFκB [112,113,114]. It is worth noting, however, that the anti-apoptotic role of CRAF does not depend on its ability to activate ERK. Additionally, targeting CRAF in KRAS G12V/Trp53 mutant lung tumors triggered massive apoptosis without blocking ERK phosphorylation, which may explain the acceptable levels of toxicity of this approach [107]. In addition to its role in regulating Fas, mitochondrial CRAF has been shown to protect the cell from apoptosis; indeed, some growth factors have been reported to play roles in the translocation of CRAF to mitochondria where p21-activated kinase (Pak1) phosphorylates it on Ser338 in the NTA motif [115,116,117]. There is also evidence suggesting that CRAF has a scaffold function for PKCθ to facilitate phosphorylation of BAD as a way of CRAF regulation [118]. In addition, CRAF directly interacts with voltage-dependent anion channels and prevents the release of cytochrome C from mitochondria [119]. Other targets of CRAF include two pro-apoptotic kinases, ASK1 [120,121,122] and MST2 [123], which also rely on a kinase-independent activity of CRAF. Deficiency of CRAF promotes apoptosis of cardiomyocyte, which can be rescued with ASK1 knockdown [124]. Despite these reports, the definitive mechanism through which mitochondrial CRAF prevents apoptosis requires further investigation. 

Similar to CRAF, ARAF also binds and sequesters MST2 independent of its kinase activity [125]. ARAF interaction with MST2 is dependent on the presence of full-length splicing product controlled by hnRNP H [126]. In primary tumors and cell lines, ARAF and MST2 co-localize to mitochondria and prevent apoptosis [126]. In addition to MST2, ARAF was found to bind pyruvate kinase M2 (PKM2), which is an embryonic splice variant of PKM and responsible for aerobic glycolysis [127]. In ARAF-transformed NIH3T3 cells, PKM2 is changed from dimeric to a highly active tetrameric conformation, linking ARAF to cancer metabolism [128]. 

Double knockout of RAF proteins generated more severe phenotype which suggested their additive effects [129,130]. Studies indicated that RAF proteins could compensate for lack of one of the isoforms up to a certain point but all three are required for regulating the development of the organism. In addition to knockouts, constitutively-active BRAFV600E mutant mouse also showed a embryonically lethal phenotype [131]. 

## 5. Activation of RAF Proteins

### 5.1. Auto-Inhibited State of RAF

Once the RAS/RAF/MEK/ERK pathway was identified, research efforts focused on studying both negative and positive regulators of the signaling cascade. In non-dividing cells, RAF exists in an auto-inhibited state where its N-terminus docks on the kinase domain, inhibiting its catalytic activity [132,133]. This is supported by the fact that overexpression of the N-terminal CR1 domain inhibits kinase activity in trans [132]. The interaction between the two parts of CRAF was further validated by the study of the N-terminus mutations of CRAF in regulating its kinase activity [132]. Moreover, RAF inhibitors that disrupt the auto-inhibited state trigger paradoxical activation, much to the disappointment of oncologists [134,135].

### 5.2. RAF Recruitment to Plasma Membrane by Activated RAS

RAS proteins are anchored on the plasma membrane through their prenylated C-terminal CAAX motif [136]. Upon GTP-loading, RAS is able to recruit RAF to the plasma membrane via its RAS-binding domain (RBD) [137], which consists of a ubiquitous fold shared by other RAS effectors such as PI3K p110 subunits and RAL guanine nucleotide dissociation stimulator (RALGDS) [69,138,139]. Single residue substitution in the RBD is sufficient to disrupt the association of RAF with RAS and abolish the activation of RAF [140]. Outside of the RBD, the Cys-rich region has been shown to form zinc coordinated structures that interact with phospholipids and facilitate membrane translocation of several kinases, including CRAF [141,142,143,144,145,146]. Furthermore, the Cys-rich region of RAF can also interact with the farnesyl group in RAS proteins [143,144,145,146,147]. In addition, the N-terminus before RBD domain (also called as N’-segment) has been shown to regulate the binding selectivity of both RAF and Ras isoforms [96,143,148]. Thus, RBD, Cys-rich region and N’-segment are involved in the plasma membrane recruitment of RAF proteins by active Ras. Despite the large amount of work and progress made in the understanding of RAS–RAF interaction, there are critical details missing on the major structural and functional interactions of these two proteins. For example, it is not clear how RAS triggers the de-phosphorylation of inhibitory Ser residues [149]. In addition, the manner through which the auto-inhibitory intramolecular interactions of RAF proteins are relieved is also not yet clear [150], although Phosphatase 2A (PP2A) [151] and PP1 [152,153] have been shown to regulate this process [150,151,152,153,154]. 

### 5.3. Dimerization Is a Key Event in RAF Activation

Despite the differences in their ability to trigger downstream ERK signaling, all RAF isoforms are activated through dimer-driven transactivation. Under physiological condition, Ras-driven activation of RAF proteins occurs on the plasma membrane where activated RAS promotes RAF dimerization, a key event to trigger the kinase activity of RAF proteins. The notion of dimerization-driven transactivation of RAF proteins arises from an early observation that artificial oligomerization of CRAF triggers its kinase activity [155,156]. The subsequent observation of RAS-induced heterodimerization of BRAF and CRAF under physiological conditions further supports the relevance of dimer formation [157,158]. The finding that kinase-dead BRAF was able to activate ERK signaling through dimerizing with and activating CRAF not only provide further support for the role of RAF dimerization [159], but raised the awareness that both catalysis-dependent and -independent functions of RAF are functionally important [158,160,161]. In addition to the BRAF–CRAF heterodimer, respective homodimers of the two isoforms, i.e., BRAF:BRAF and CRAF:CRAF, have also been detected, but were noted to have lower kinase activity [159]. ARAF, however, has the lowest affinity for dimer formation due to its different structural features, most notably its non-canonical APE motif that does not stabilize the dimer interface as BRAF and CRAF are able to do [84]. However, site-specific mutagenesis of this motif from AAE to APE enables ARAF to form dimers as strongly as CRAF. In summary, RAF family members can form physiologically relevant heterodimers and homodimers, resulting in their transactivation [88,162] (Figure 2b). 

The dimeric structure of RAF stabilizes the closed conformation of the kinase domain by limiting the oscillating motion of the two kinase lobes and promoting key conformational transitions [158]. Conversely, the acquisition of the active conformation also facilitates RAF dimerization. Once RAF achieves active conformation, its dimer interface becomes further stabilized by the hydrophobic R-spine residue in the αC- helix (L505 for B-Raf, L397 for C-Raf, and L358 for A-Raf) located adjacent to the conserved RKTR motif, which is allosterically connected αC-helix and dimer interface [163]. Upon the relocation of R509 to the center of the dimer interface, αC-helix interacts with the NTA motif of the trans RAF molecule and adopts the “IN” conformation [80,164] 

The dimerization of RAF proteins can be promoted by shortening their β3-αC loop; and in-frame deletions of β3-αC loop activate ARAF by enforcing homodimer formation, showing that ARAF can function as a “true” kinase to induce ERK phosphorylation [84], even though it is the weakest kinase in the family. The corresponding deletions in BRAF also ramp up its kinase activity through enhancing its homodimer formation. A surprising finding from these studies is that the kinase-dead BRAF mutant with an in-frame deletion of β3-αC loop (ΔNVTAP/V471F) can activate ERK signaling in the absence of active RAS [84] due to its substantial dimer affinity.

The dimerization of RAF proteins can be also improved in other ways. The BRAF splicing variants lacked exon 4–7 exhibit a stronger dimer affinity and thereby a strong resistance to RAF inhibitors [165]. In addition to alternative splicing products; gene fusions, translocations and deletions that remove the auto-inhibitory N-terminus allow RAFs to homodimerize with higher affinity [166,167,168,169]. Furthermore, RAF inhibitors, especially the first-generation drugs, enhance RAF dimerization upon their binding, and thereby paradoxically activate the pathway, which is further discussed below. Understanding the complete mechanism of dimerization is still a work in progress, with questions remaining about the scaffolding proteins, activation direction, and the priming of the cells for RTK signaling by increasing RAS nanoclusters. 

### 5.4. The Role of NTA Motif and Activation Loop Phosphorylation in RAF Activation

RAF proteins undergo multiple phosphorylation events during their activation cycle. Two major phosphorylation events, the NTA motif and the activation loop, play key roles. The NTA motif contains divergent loci for phosphorylation among RAFs, allowing them to have distinct regulations [75]. In BRAF, the NTA motif contains SSDD, in which the D448 and D449 provide the initial negative charge before serine phosphorylation. In CRAF, this locus contains SSYY and requires phosphorylation of both serine and tyrosine residues. The constitutively phosphorylated SS and acidic DD in the NTA motif of BRAF can explain the higher activity of BRAF compared to ARAF and CRAF. Negative charges in this loci contribute to the relief of autoinhibition [132,150,170] and are critical for dimerization-driven transactivation [80]. In RAF homo/heterodimers, the phosphorylated NTA motif was suggested by molecular modeling to form multiple salt bridges that extend and stabilize the dimer interface between two protomers [116,129]. It has been shown that the phosphorylation status of NTA motif dictates the direction of transactivation in RAF dimers [80]; the protomer with phosphorylated NTA motif acts as an activator, while the other protomer with non-phosphorylated NTA motif does as a receiver in RAF dimers. A ‘receiver’ protomer can be switched into an ‘activator’ protomer upon phosphorylation in its NTA and thereby amplify the dimerization-driven transactivation of RAF proteins. Protein kinases that target the NTA motif play an important role not only in the activation of RAF proteins but also by controlling the receiver–activator switch. SRC family kinases are the main kinase family targeting YY in ARAF and CRAF [171,172,173,174,175]. Although still controversial [176], p21-activated kinase (Pak) family members acting downstream of PI3K-CDC42 or RAC signaling cascades have been suggested to phosphorylate Ser338 [177,178] in CRAF. Other kinases that potentially target the NTA motif of RAF proteins include the following: (1) Janus Kinase 2 (JAK2), which is able to phosphorylate CRAF to activate MEK1 [179], (2) Casein Kinase 2 (CK2), which phosphorylates CRAF at Ser338 and BRAF at Ser446 [180], and (3) Calcium/calmodulin-dependent protein kinase II (CaMKII), which can phosphorylate CRAF at S338 [181].

The phosphorylation of activation loop is also important for the function of RAFs. The dimerization of RAF proteins facilitates their transition to an active conformation, which directly induces their activation loop auto-phosphorylation as a consequence. Like most protein kinases, the activation loop of RAFs contain two conserved phosphorylation sites relevant to their kinase function; for ARAF, these are Thr452 and Thr455 [182]; for BRAF, these are Thr599 and Ser602 [183]; and for CRAF, these are Thr491 and Ser494 [184]. Although studying the phosphorylation of the activation loop is difficult due to highly dynamic de-phosphorylation, the data from RAF co-activation assay suggest that cis auto-phosphorylation is the mechanism [80]. A recent study of a mouse model with BRAF T599A/S602A mutation [185,186] showed that the loss of activation loop phosphorylation could not duplicate the lethal phenotype of BRAF null mice. However, these mice had an aberrant development of the hematopoietic system and reduction of p-ERK level in the brain, among other characteristics, indicating the importance of proper activation loop phosphorylation of BRAF in development, function, and maintenance of cell populations [187]. 

## 6. Regulation of RAF by Accessory Molecules

Activation of RAF is also regulated by other proteins, including heat-shock protein 90(Hsp90) [188,189], CDC37 [188], kinase suppressor of Ras (KSR), and 14-3-3 [190,191,192,193] proteins. 

### 6.1. Hsp90/Cdc37 Chaperone Complex

The Hsp90/Cdc37 chaperone complex participates in proper protein folding that stabilizes protein kinases, including RAFs [194]. The association of RAF proteins with hsp90/cdc37 complex is essential for their activity towards MEK. Further, the association of BRAF with hsp90/cdc37 complex facilitates the assembly of high molecular weight BRAF complex and promotes its kinase activity, probably by increasing dimer affinity [195]. It is not surprising, therefore, that Hsp90 inhibitors can block the activity of RAF proteins and also induce their degradation, especially that of the constitutively active RAF mutants such as BRAF(V600E) [188,196,197]. Recently, Hsp90 inhibitors were shown to prevent development of resistance to RAF inhibitors on BRAF(V600E)-harboring cancers in clinical trials, even though the underlying mechanism were too complicated to pinpoint a single molecule [198,199,200]. 

### 6.2. KSR

Kinase suppressor of RAS (KSR) was initially identified through screening molecules essential for Ras function in Drosophila [201]. It had been referred to as pseudo-kinase due to its low kinase activity [202]. Subsequently, KSR was identified as a scaffold protein that not only brings close proximity between RAF and MEK1 [203], but also allosterically activates RAF [158,204,205,206]. It should be noted that KSR forms not only a side-to-side dimer with RAF proteins but also a face-to-face dimer with MEK, both of which are critical for its ability to transactivate RAF proteins [158,207,208]. 

### 6.3. Proteins 14-3-3

The 14-3-3 proteins were the first identified, and the most well-known, phosphoserine/phosphothreonine binding proteins, which can interact with a wide variety of proteins including transcriptions factors, cytoskeletal proteins, apoptosis factors and tumor suppressors. Binding to 14-3-3 can alter the proteins’ stability, localization, conformation, and association with other proteins. The functions of 14-3-3 has been reviewed extensively [209,210]. 

The 14-3-3 binds to phospho-serine/threonine residues in two conserved motifs of RAF proteins: RSXpS/TXP or RXXXpSXP [78,211]. RAF proteins contain two 14-3-3 binding sites: S259 and S621 for CRAF [79,212]; S365 and S729 for BRAF; and S214 and S582 for ARAF (Figure 2a). The 14-3-3 association with these two sites plays opposite roles in RAF activity. Activation of RAF by 14-3-3 occurs in the event of 14-3-3 binding two RAF molecules at the C-terminal phosphoserine, which promotes dimerization. Cryo-EM studies have shown that a dimeric 14-3-3 binds two phosphorylated serine residues on different RAF proteins, such as CRAF at Ser621 and BRAF at Ser 729, and thereby stabilizes the side-to-side heterodimer or homodimer [77,213]. On the other hand, if a dimeric 14-3-3 binds to the N- and C-terminus of a single RAF, respectively, it can stabilize autoinhibitory conformation [211,213]. The serine residues in conserved 14-3-3 binding motifs can be phosphorylated by Protein Kinase A (PKA) [214,215,216], Akt [216,217,218], AMPK [219,220], or by LATS1 from the MST2-Hippo pathway [221]. Regulation of phosphorylation events at two 14-3-3 binding sites can change the response drastically due to their opposite activity. The 14-3-3 binding site phosphorylation by different kinases may influence the therapeutic efficacy of cancer drugs. For example, in Ras-mutated cancer cells, the CRAF S621 is phosphorylated redundantly by AMPK and itself. Combination of RAF inhibitors with AMPK inhibitor could reduce the paradoxical activation [219]. Recognition of the alternative kinases that could phosphorylate 14-3-3 binding sites and thereby alter RAF activity, could improve clinical success. 

## 7. RAF Function as a Dimer

As mentioned above, the Arg at the center of RAF dimer interface is a key residue for RAF function, and its mutation to His blocks the dimerization-driven transactivation of RAF proteins [158]. This mutation can also abolish the drug resistance of BRAF(V600E) splicing variants that lack a part of N-terminus and thereby have a higher dimer affinity than their prototype [165]. Based on these observations, a monomer hypothesis in which RAF inhibitors bind and inhibit monomeric BRAF(V600E), but not dimeric variants, has been suggested to explain the drug resistance of BRAF mutants. However, this hypothesis has been challenged by other findings [80,84,95,195,222,223]. Firstly, it was shown that BRAF(V600E) has an extended dimer interface in contrast to its wild-type counterpart and exists as dimer/oligomer when expressed in cells [195,223]. Secondly, the Arg-to-His mutation did not fully diminish dimer formation of RAF proteins [223], and some RAF mutants with high dimer affinity such as BRAF(ΔNVTAP) and BRAF(ΔMLN) were still able to transactivate wild-type RAF in the presence of Arg-to-His mutation [84]. Thirdly, the Arg-to-His mutation, together with APE motif alteration, completely dissociated BRAF(V600E) dimers and abolished their activity, which could be rescued by GST fusion-mediated re-dimerization [84]. Lastly, RAF had been shown to phosphorylate MEK in a dimer-to-dimer manner in which an active RAF needs the other RAF partner to facilitate MEK phosphorylation (further discussed below) [84]. Therefore, the active monomer hypothesis is not supported at its current standing. 

## 8. RAF–MEK Heterodimerization and MEK–MEK Homodimerization, Essential Events for Signaling

MEK was identified independently by multiple groups as a substrate of RAF in 1992 [19,224,225], and their interaction was further supported by the yeast two-hybrid assay [226]. Following these findings, MEK was shown to be phosphorylated by RAF on Ser218 and Ser222 in its activation loop [227]. Recent studies have provided detailed mechanisms of RAF phosphorylation and activation of MEK. As a substrate, MEK needs to be recruited to RAF before activation. In quiescent cells, BRAF and MEK form a heterodimer in the cytosol, while CRAF and ARAF do not interact with MEK under this condition [86], presenting the question of how MEK is recruited to these RAFs. Crystallography studies have revealed that BRAF interacts with MEK1 in a face-to-face manner through two contact sites [86,208]. The first contact site is αG helices, a structural component of kinase–substrate docking interaction. The second contact site is consisting of their activation loop which generates antiparallel β-sheet [86,208]. Mutations on both contact sites that disrupt the BRAF/MEK interaction block both allosteric and catalytic activities of BRAF [84], suggesting that the RAF/MEK association plays an indispensable role in signal transmission from RAF to MEK. 

Given the abilities of RAF proteins to form side-to-side dimers with themselves and face-to-face dimers with MEK, it is not surprising that RAF and MEK assemble a tetramer complex of MEK:RAF:RAF:MEK in the process of activation, which has been captured in crystal structures [86]. Although how the RAF dimer phosphorylates MEK in this transient RAF/MEK tetramer is not completely understood, recent studies suggested that two MEK molecules need form a homodimer that is further phosphorylated by RAF dimer or itself [84,222], since monomeric MEK cannot be phosphorylated by RAF, and MEK homodimerization drives autophosphorylation. Moreover, phosphorylated MEK exerts its activity towards ERK as a dimer [222], suggesting that MEK/MEK homodimerization plays a critical role, as do RAF/RAF and RAF/MEK dimerizations, in the pathway activation.

## 9. Feedback Inhibition and Return to the Inactive State

Cessation of the activated ERK signal is a crucial part of controlled cell division; therefore, the triggered ERK pathway needs to return to the basal state with regulated feedback cues. As a part of immediate inhibitory phosphorylation; SOS1, the RAS-GEF, is phosphorylated by ERK, which inhibits interaction with Grb2 and creates 14-3-3 binding site for inhibitory binding [228,229]. ERK also exerts negative feedback on RAF by phosphorylating multiple Ser/Thr sites on RAF [230,231,232]. ERK phosphorylation of these Ser/Thr sites breaks the interaction of RAF with Ras, and also RAF-RAF dimerization [229]. Additionally, some of these Ser/Thr sites can also be phosphorylated by c-Jun amino-terminal kinases (JNKs) as a protective mechanism under stress conditions [233]. Further, the autophosphorylation of P-loop residues in RAF proteins serves as another feedback mechanism that impairs their kinase activity [234]. For MEK, it can be phosphorylated at its N-terminus that intercepts its activity [235].

Shortly after inhibitory phosphorylation as described above, the components of Ras/RAF/MEK/ERK signaling cascade require dephosphorylation to return to the inactive state. It has been shown that PP5 participates in the dephosphorylation of pSer338 in the NTA motif of CRAF [236], while PP2A is involved in Ser/Thr site dephosphorylation in a PIN1 dependent manner [231]. However, in the area of dephosphorylation, most underlying molecular mechanisms remain to be elucidated. 

Outside of the signaling cascade and immediate inhibitory phosphorylation, Dual-specificity phosphatases (DUSP) and Sprouty proteins are involved in transcriptional inhibition of the ERK signaling [237]. DUSP6 is upregulated by its transcription factor that activated by ERK1/2 [238,239]. Sprouty proteins are induced upon growth factor treatment [240], which can inhibit ERK signaling through different mechanisms. Spry1 and Spry2 are phosphorylated at their N-terminus upon RTK activation, which enables Spry to dock on Grb2 [241]. Furthermore, Spry4 inhibits ERK signaling by binding to CRAF [242]. 

## 10. Mutations in the Ras/RAF/MEK/ERK Signaling Cascade

The RAS signaling cascade has well-defined role in tumorigenesis. Point mutations across each of the members within the cascade have been identified as either driver of the tumor formation (RAS and RAF mutations) or indicator of poor prognosis (MEK and ERK mutations). Interestingly, driver mutations within the pathway are mutually exclusive. In this section, we describe the mutation hotspots for each member of the signaling cascade.

### 10.1. Ras Mutations

RAS mutations in all cancer types occur mostly in KRAS (85%), followed by NRAS (12%) and HRAS (3%). Interestingly, mutants of different RAS isoforms have tissue/cancer-type-dependent distributions. For example, NRAS is the most common in melanoma, while HRAS in adrenal glands and KRAS in pancreas [243]. Moreover, KRAS G12D was shown to promote stronger colon cancer development than NRAS G12D in Apc-deficient mice [244], and HRAS G12V knock into the KRAS locus was not tumorigenic in the lungs of mice [245]. 

RAS proteins have two mutational hotspots, 12–13 and 61. Most mutations changing G12 or G13, likely to intercept GAP’s Arg finger loop accession to the RAS GTPase site and prevent it from promoting hydrolysis [246,247,248]. As a consequence, G12 and G13 mutants trap RAS in a constitutively active state. Mutations at Q61 inhibits intrinsic GTP hydrolysis and GAP-mediated GTP hydrolysis (Figure 3) [246]. It is worth noting that, in addition to the isoform-dependent tissue distribution in RAS-driven cancers, the site of mutation hotspots also comes into play in the tissue/cancer-type distribution. For example, 90% of KRAS mutations in pancreatic ductal adenocarcinoma are at the position G12, while 90% of NRAS mutations in melanoma are at Q61. These patterns might indicate underlying fundamental signaling landscapes and RAS mutant interplay with these landscapes. Adding to the complexity of RAS mutations, the oncogenic stimuli also mold the site of mutation and the type of tumor development. For example, KRAS Q61R/L dominates the G12 mutation in urethane treated mice [31]. 

### 10.2. RAF Mutations

Although V-RAF was discovered as a first oncogenic Ser/Thr protein kinase, its cellular prototype CRAF is rarely mutated in cancers. The function of RAF as a prominent cancer driver was not established until the discovery of BRAF(V600E) in 2002 [26,249]. In cancer genomes, BRAF is a major target of oncogenic mutations and a single-point mutation, and V600E represents >90% of events, while mutations in CRAF, ARAF, and KSRs are much less and have been detected in decreasing order of frequency (Figure 4) [250]. 

Cancer-related mutations of RAF are enriched in special domains of the proteins and can be categorized into multiple groups based on how they trigger the pathway. The first group (or Class I) of RAF mutations activate RAF by mimicking phosphorylation of the activation loop. The second group (or Class II) of RAF mutations turn on RAF activity by relieving the auto-inhibitory status. The third group (or Class III) of RAF mutations have no, or impaired, kinase activity and agonist the pathway through transactivating their wild-type counterparts. 

Class I RAF mutations include V600E/D mutations in activation loop of BRAF and constitute the largest group in the RAF mutation spectrum [251]. The V600E/D mutation stabilizes the active conformation of BRAF by forming a salt bridge with K507 [252,253], thereby dramatically triggering the kinase activity of BRAF independent of Ras [157,165,223,231]. Class II RAF mutations are mainly located at the activation loop (K601 [254] and L597 [255,256]), or Gly-rich loop (P-loop, G464 [257] and G469 [258]), and disrupt the inhibitory interaction of activation loop with Gly-rich loop and thus destabilize the auto-inhibitory status [253]. This type of mutant has intermediate kinase activity and increased dimer affinity, and it triggers ERK signaling with or without active RAS. Class III RAF mutations are mostly found in the Gly-rich loop (G466 [257,259] and G469E [260]), the DFG motif (D594 [254,261] and G596 [254,259]), the catalytic loop (N581 [252]), or the C-spine (V471F [91]). These mutants have greatly reduced kinase activity compared with wild-type RAF and drive the activation of ERK signaling by transactivating the wild-type RAF with their enhanced dimerization affinity [160,252,262]. Unlike the Class I and Class II mutants, the Class III mutants require active Ras to trigger signaling cascade [80,262].

In addition to the highly prevalent mutations above, there are also some minor populations of RAF mutations in cancer genomes. N-terminal truncations of CR1 and CR2 domains, or kinase domain fusions with other proteins that relieve the N-terminal inhibition, have been identified in RAF genes [167,169,263,264,265]. These N-terminal truncations or kinase domain fusions allow RAF to dimerize and activate ERK signaling in a RAS-independent manner. Under physiological conditions, RAF dimerization occurs on the plasma membrane. The mutations in the Cysteine-rich domain and Ras-binding domain (RBD) of RAF can promote its plasma membrane recruitment, and thereby dimerization, to trigger ERK signaling [250,266,267,268,269]. In addition, mutations that abolish the inhibitory 14-3-3 association with the N-terminus of RAF (S259 in CRAF and S365 in BRAF) were also found in a small group of cancers [149,186,270,271].

### 10.3. MEK and ERK Mutations

In contrast to Ras and RAF mutations, MEK mutations are much less common in cancer genomes. These mutations do not co-occur with Ras or RAF mutations, indicating that they function as cancer drivers [272]. MEK mutations are classified into two groups according to their activation mechanism. The first group of MEK mutations turns on the kinase activity of MEK by disrupting the inhibitory intramolecular interaction mediated by the regulatory helix A, while the second group does so through enhancing MEK homodimerization. These two types of MEK mutants also exhibit differential sensitivities to MEK inhibitors in clinic or under clinic trials [222]. Like RAF, the elevated dimer affinity may also result in the resistance to inhibitors. Finally, ERK mutations are very rare in cancer genomes. However, a mutational hotspot has been identified on the site of D321/E322, which blocks the Dusp6-mediated dephosphorylation of ERK and thereby extends the half-life of active ERK [273].

## 11. Targeted Therapies against Hyperactive Ras/RAF/MEK/ERK Signaling in Cancers: The Present State and Perspectives

### 11.1. Is RAS Druggable?

Oncogenic Ras mutants have been considered “undruggable” over the decades due to their high affinity with GTP and the lack of proper binding pocket for small molecule inhibitor binding [274]. Recently, the Shokat group at UCSF showed that KRAS G12C could be targeted by using a covalent small molecule that docks in the switch II pocket and cross-links with Cys12 [275]. This discovery spurred the race to develop KRAS-G12C-targeting drugs for clinical use. AMG510 and MRTX1257 were the first ones to be developed as drugs to target KRAS G12C mutant for non-small cell lung cancer [274,276,277,278]. The results from phase I trials were promising and showed a 50% response rate for patients with KRAS G12C mutations. However, these drugs still require further investigation; besides validating efficacy, the impact of co-mutations and the development of drug resistance also need to be evaluated [276]. Unfortunately, KRAS G12C is only represented in a small fraction of RAS mutant cancers, and the challenge remains to drug the other RAS mutants. 

Among alternatives to target mutant RAS isoforms, inhibiting their functionally relevant post-translational modification represents a promising approach. The protein prenylation processing pathway attracted significant attention in the early targeted therapeutic field [48,279]. Initial effort was largely focused on protein farnesyltransferase, for good reason [48,279]. First, it was discovered as the canonical prenylation enzyme for all RAS isoforms under normal cellular conditions [44,51]. Second, it has been under careful genetic, biochemical, structural and functional studies [280,281,282,283]. However, as the first targeted therapy development, inadequate understanding of the alternative prenylation pathway of K- and N-RAS by geranylgeranyl transferase and corresponding inadequate stratification method led to the dismal clinical results for farnesyltransferase inhibitors. In recent years, new trials have been initiated that are designed to target the HRAS driven cancers, as H-RAS is not subject to alternate prenylation [284]. It will be interesting to see the level of efficacy from these trials; early data has shown promise (https://www.aacr.org/Newsroom/Pages/News-Release-Detail.aspx?ItemID=1350).

The earlier dismal trial results for farnesyltransferase inhibitors also sparked the interest in protein geranylgeranyl transferase (GGTase I), which prenylates K- and NRAS when farnesyltransferase is inhibited [284,285]. Targeting GGTase I, alone or in combination with FTIs, became an active area of investigation following the FTI trials [286]. Indeed, there are major efforts in developing GGTase I inhibitors for the treatment of certain cancers [286,287,288]. However, the toxicity of combination therapy of GGTase I and FTase may have limited the pace of such development [289]. Among the three steps of prenylation modifications, the last step of carboxymethylation by isoprenylcysteine carboxylmethyltransferase (ICMT) has also attracted much attention. Genetic suppression of ICMT in both cell and mouse models showed that ICMT inhibition led to the inhibition of tumorigenesis in multiple cancer model systems [55,290]. Proof-of-concept ICMT inhibitors have largely replicated the efficacy of genetic cancer models [291,292,293]. It will be quite interesting to follow the progress and evolvement of the development. 

For some RAS mutants, targeting their key effectors could be an effective alternative method. The small molecule inhibitor Rigosertib has been used to block the interactions of active Ras with the Ras-binding domain of effectors, and it has shown efficacy to target Ras-mutated cancers in phase III clinic trials [294]. Sulindac and MCP110, two other Ras-effector interaction inhibitors, were also shown to inhibit RAS driven tumorigenesis [295,296,297].

Like other oncogenes, Ras mutants need a set of cellular factors to facilitate their oncogenicity. These synthetic lethal factors/networks could be targeted for cancer therapy. Two recent studies showed that targeting RAF/MEK/ERK signaling together with autophagy signaling led to synthetic lethality and achieved a promising efficacy on Ras-driven cancers [298,299], suggesting that synthetic lethal factors/networks may be highly valuable for developing new drugs/methods against Ras-mutated cancers. Indeed, prior evidence of manipulating autophagy by inhibiting ICMT has shown encouraging results [300,301,302], which supports that ICMT can be used to regulate RAS-driven signaling. For the effort of identifying the targets downstream of ERK signaling in modulating autophagy, several screens have been carried out with whole genome shRNA knockdown libraries, which yielded few druggable factors [303]. The strong biology, however, appeals to future reliable screening approaches such as Cas9-CRISPR-mediated knockout [304]. Nevertheless, exploring all synthetic lethal factors/networks of Ras should provide better understanding how oncogenic Ras induces cancers and also accelerate the development of drugs against Ras-mutated cancers.

### 11.2. RAF/MEK/ERK Inhibitors and Resistance

Among the members of the Ras/RAF/MEK/ERK signaling cascades, RAF is a key direct effector of oncogenic Ras mutants and also a prominent target of oncogenic mutations. As the first kinase in this pathway, RAF has been thought as an ideal target for drug development against cancers. The first-generation RAF inhibitors, Vemurafenib [305,306], Dabrafenib [307], and Encorafenib [308], were developed and applied to treatment of BRAF(V600E)-harboring cancers as single agents or together with MEK inhibitors. These drugs achieved a promising efficacy at the initial therapeutic phase, although this efficacy was abrogated by quick-rising drug resistance. Mechanistic studies have shown that cancer cells reactivate this pathway upon drug treatment through two different ways: (1) upregulating the cellular level of active Ras, which leads to paradoxical activation of ERK signaling; and (2) alternative splicing of BRAF(V600E) to generate variants with truncated N-terminus, which enhances BRAF(V600E) homodimerization and decreases drug affinity. Interestingly, these drug-resistance cancer cells become addicted to RAF inhibitors, and drug withdrawal delays the growth of resistant cancers (Figure 5). This phenomenon can be explained by the concept of a “sweet-spot” for hyperactive Ras/RAF/MEK/ERK-signaling-driven cancer progression (Figure 5a). Specifically, cancer cells need optimal ERK signaling for growth, and ERK signaling that is too high will induce cell death or senescence and hence be toxic to cancer cells. Drug-resistant cancer cells have much higher ERK signaling than drug-sensitive cancer cells. Drug treatment decreases the ERK signaling in drug-resistance cancer cells to a level suitable for growth, while a drug withdrawal or “drug holiday” will inhibit their growth [309,310] (Figure 5b,c).

The paradoxical effect of the first-generation RAF inhibitors not only abrogated their efficacy but also induced secondary malignancies, a major side-effect of RAF inhibitors. To overcome the drug resistance of the first-generation RAF inhibitors, second-generation of RAF inhibitors were developed and underwent clinical trials; such inhibitors include pan-RAF inhibitors (such as LY3009120 [311], TAK632 [312], TAK580 [313], CCT3833 [314], BGB283 [262], BAL3833 [315], LXH254 [316], and RAF265 [317]), and paradox breakers (such as PLX8349 [318,319,320]). The pan-RAF inhibitors can inhibit both protomers in RAF dimers with similar affinity, while the paradox breakers induce an αC helix-out conformation upon loading and thereby prevent dimerization-driven transactivation.

To block hyperactive Ras/RAF/MEK/ERK signaling in cancers, MEK and ERK have also been used as targets for drug designs. Two MEK inhibitors (trametinib and cobimetinib) have been developed and approved for treating BRAF(V600E)-harboring cancers as single agents or together with RAF inhibitors, while ERK inhibitors are still undergoing clinical trials. In contrast to RAF inhibitors, these inhibitors have no paradoxical effect, but they do have a lower therapeutic index since they strongly inhibit this signaling pathway in normal cells [321,322]. This also implies that targeting downstream MEK/ERK may not be a good choice to treat Ras- or RAF-mutated cancers.

The development of RAF/MEK/ERK inhibitors has significantly improved targeted cancer therapy, and it has also accelerated our understanding of molecular mechanisms that tune the output of Ras signaling, which in turn facilitates the development of next-generation inhibitors. Studies on these inhibitors have revealed a dominant role of component interactions that include RAF/RAF, RAF/MEK, and MEK/MEK dimerizations in regulating Ras signaling. Since the structures of these complexes have been resolved, the development of allosteric inhibitors that break these interactions would be an attractive area of research in the coming years. In contrast to the first- and second-generation RAF inhibitors, these allosteric inhibitors should have many more advantages, such as no paradoxical effect and less off-target effects. 

## 12. Closing Remarks

The second-generation RAF inhibitors, which are expected to show longer-lasting effects on cancer growth, are now entering the clinical market [323]. Despite the enthusiasm, it is still not well understood how well cancers will respond to these drugs in the long term [262,324]. The studies on inhibitor-RAF interactions and the wild-type RAF response to the new inhibitors could help minimizing the off-target effects in patients. Determining the crystal structure and direct interaction between inhibitors/effectors would provide better insights into their mode of action in the pathways. This information not only could create potentially avenues for improved inhibition with better the drug specificity and efficacy; it could also reveal and validate novel allosteric targets within the pathway. Understanding the interactions between RAS/RAF/MEK mutants and their wild-type counterparts could result in the discovery of novel allosteric target sites to block the formation of dimers and tetramers between mutant and wild-type proteins. Besides the focus on RAS-downstream-signaling-partner studies, it is also interesting to follow the progress in the effort of targeting RAS functions, either directly or via modulating the functional post-translational prenylation pathway enzymes. With these comprehensive research efforts, we hope to have more specific and potent drugs targeting RAS/RAF/MEK/ERK signaling, to aid in the management of human cancers.

## Figures and Tables

**Figure 1 cells-09-00198-f001:**
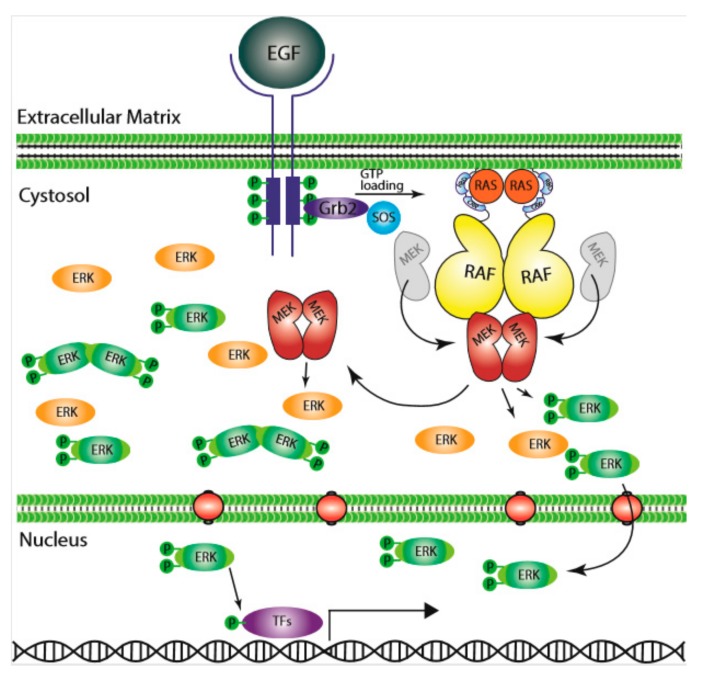
The Ras/RAF/MEK/ERK signaling pathway. Epidermal growth factor (EGF) initiates the signal on the cell surface through the EGF receptor (EGFR) (receptor tyrosine kinase), which activates guanine exchange factor to load RAS with GTP. RAS–GTP dimers/nanoclusters recruit RAFs or RAF/MEK heterodimers to plasma membranes, where RAF and MEK assemble transient tetramers that facilitate RAF activation through a back-to-back dimerization. MEKs docking on active RAF dimers further form face-to-face homodimers that are turned on by RAF. Activated MEKs phosphorylate ERKs, which generate response to the signal. CRR; Cys-rich region, RBD; Ras-binding domain.

**Figure 2 cells-09-00198-f002:**
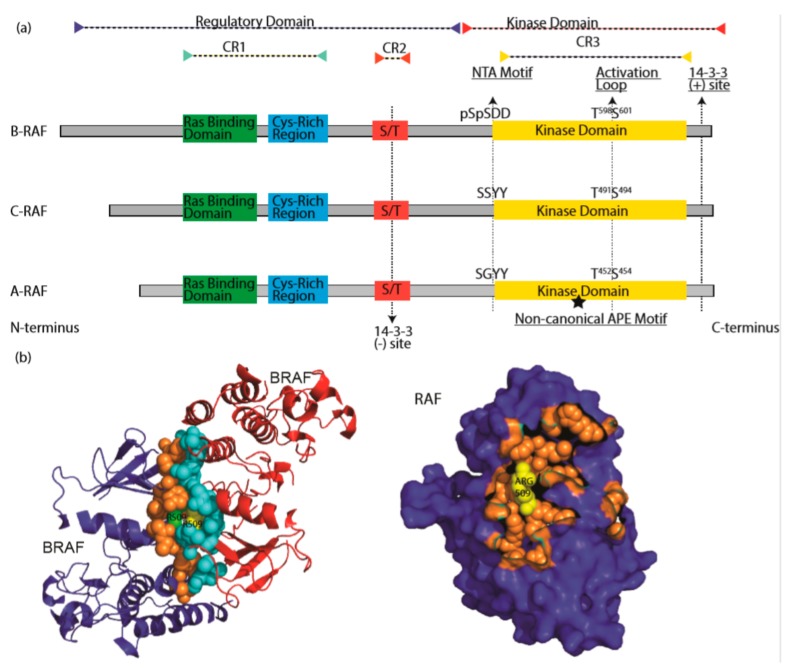
The RAF family kinases. (**a**) Conserved domains on three RAF proteins are shown in panel (**a**). CR1 contains a Ras binding domain and a Cys-rich region while CR2 includes a S/T phosphorylation site. The 14-3-3 binding at this region inhibits RAF. CR3 contains a putative kinase domain adjacent to an acidic N-terminus (NTA) and a regulatory C-terminus. At the C terminus, there is a secondary 14-3-3 binding site which promotes dimerization. The non-canonical APE motif of ARAF is labeled with a star. ((**b**), left) Dimer interface of BRAF is shown, crystallography data was obtained from [86], PDB ID: 4MNE. Blue and red color indicates two separate BRAF molecules. Orange (Blue BRAF) and turquoise colored (Red BRAF) sphere-shaped amino acids indicate the dimer interface. While R509 from both RAF molecule located at the center of the dimer interface, green R509 belongs to red BRAF, while yellow R509 does to blue BRAF. ((**b**), right) Crystal structure of dimer interface from plane of interaction with only blue BRAF chain is visible. Orange spheres indicate the amino acids at the dimer interface, with exception of Arg509, which is labeled with yellow color. Structure was drawn by using pyMOL. CR1: conserved region 1; CR2: conserved region 2; CR3: conserved region 3.

**Figure 3 cells-09-00198-f003:**
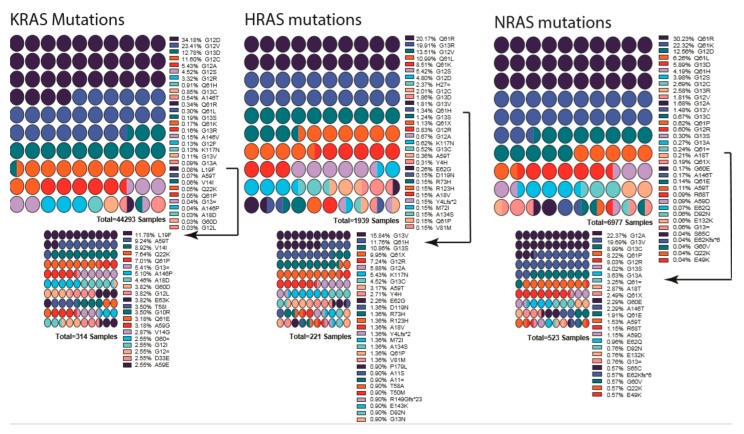
List of mutations detected in each RAS gene isoform. Percentages are indicated next to the mutation, and colors indicate the mutation. Low-percentage mutations were shown as a smaller graph underneath due to their being almost invisible in first graph. Arrows indicate from which point onward graphs were cut into two. Color scheme repeats every ten mutations and should be interpreted in combination of percentages and order. Graphs were drawn by using Prism 8.

**Figure 4 cells-09-00198-f004:**
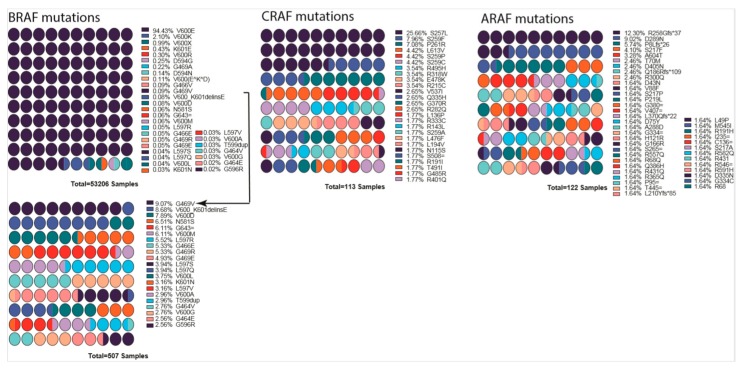
List of mutations detected in each RAF gene isoform. Percentages are indicated next to the mutation, and colors indicate the mutation. BRAF low percentage mutations are shown as a smaller graph underneath due to their being almost invisible in the first graph. Arrow indicates from which point onward graph was cut into two. Mutation data were obtained from COSMIC database. Color scheme repeats every ten mutations and should be interpreted in combination of percentages and order. Graphs were drawn by using Prism 8.

**Figure 5 cells-09-00198-f005:**
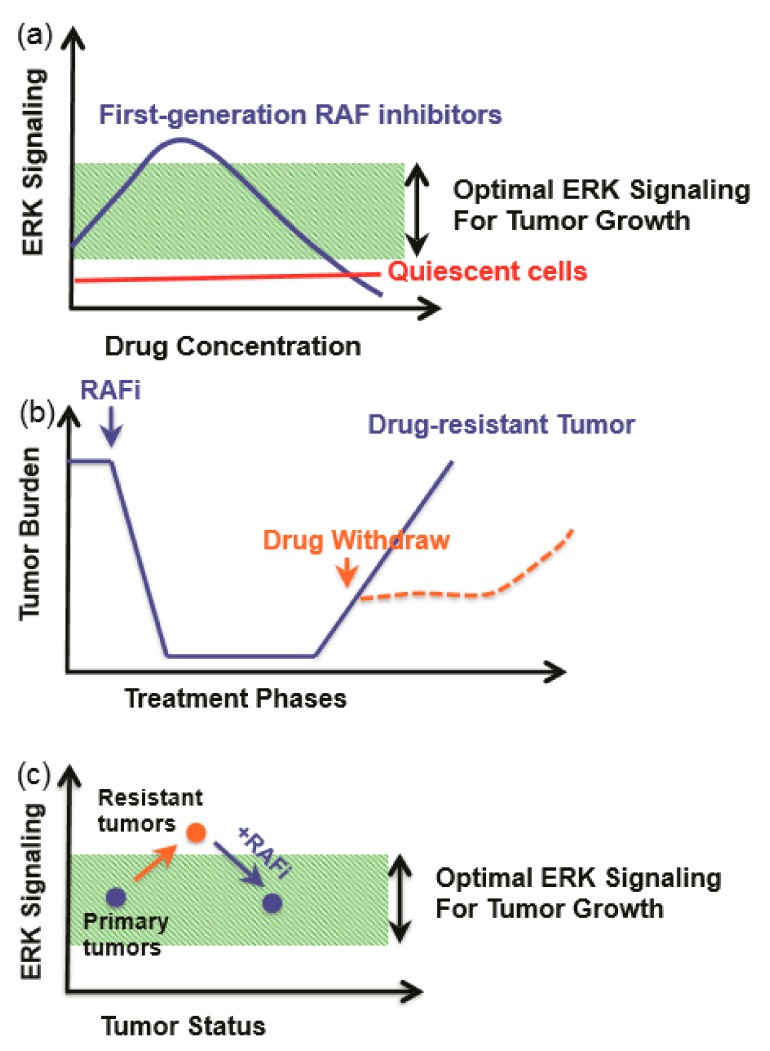
An optimal ERK signaling is required for tumor growth. (**a**) First-generation RAF inhibitors paradoxically agonist ERK signaling in cancer cells with active Ras. (**b**) Drug-resistant tumors rely on the presence of the drug for optimal growth, and a removal of the drug delays tumor progression. (**c**) ERK signaling levels in response to phenomenon in panel (**b**). RAF inhibitor keeps ERK signaling within optimal zone.

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
