# Peer review of "Targeting Aberrant RAS/RAF/MEK/ERK Signaling for Cancer Therapy"

_cells, 2020, doi:10.3390/cells9010198_

Round 1

Reviewer 1 Report

The review submitted by Degirmenci et al., entitled "Targeting aberrant RAS/RAF/MEK/ERK signaling for cancer therapy" is comprehensive, insightful, and interesting manuscript. I have only a few suggestions that may help in improving this manuscript.

Minor suggestions:

Figure 1: In this schematic, RAF is not interacting with the membrane. Considering RAF recruitment to the membrane is essential for its activation, it would be good to show them close to the membrane, primarily the Cys-rick domain present in the CR1. Line 477, it is correct that any mutations at G12, except to Pro, results in impaired GAP-mediated GTPase activity. However, this is not true for G13 mutants. Mutation at G13 to residues with smaller side chains (G13D, G13C, G13A, etc.) still show neurofibromin-mediated GAP activity. (PMID: 18713003 and 31611389)  Recent cryo-EM publications on active and inactive complexes of RAF have provided new insights into the RAF biology. It may be useful to include a brief update on these findings. It would be useful to include a couple of sentences on the N-terminal differences in RAF proteins (the region before RBD) and possible roles they may play in RAS/RAF biology.

Author Response

Reviewer #1

The review submitted by Degirmenci et al., entitled "Targeting aberrant RAS/RAF/MEK /ERK signaling for cancer therapy" is comprehensive, insightful, and interesting manuscript. I have only a few suggestions that may help in improving this manuscript.

We thank the reviewer #1’s positive comments to our manuscript, and we have revised our manuscript according to his/her suggestions. We believe the reviewer #1 would be satisfied with the new version of our manuscript.

Minor suggestions:

Figure 1: In this schematic, RAF is not interacting with the membrane. Considering RAF recruitment to the membrane is essential for its activation, it would be good to show them close to the membrane, primarily the Cys-rick domain present in the CR1.

Thanks for reviewer’s good suggestion. We have revised the figure 1 accordingly. In the new figure, the Ras binding domain and the Cys-rich region are labelled as RBD and CRR respectively, and also shown close to the plasma membrane.

Line 477, it is correct that any mutations at G12, except to Pro, results in impaired GAP-mediated GTPase activity. However, this is not true for G13 mutants. Mutation at G13 to residues with smaller side chains (G13D, G13C, G13A, etc.) still show neurofibromin-mediated GAP activity. (PMID: 18713003 and 31611389) 

We thank the reviewer’s kind reminding of this issue. Since we do not intend to discuss the difference among variable G12/13 mutations in detail, we have changed it to a general description as “Most mutations changing G12 or G13, likely intercept GAP’s Arg finger loop accession to the RAS GTPase site and prevent it from promoting hydrolysis…”

Recent cryo-EM publications on active and inactive complexes of RAF have provided new insights into the RAF biology. It may be useful to include a brief update on these findings.

We totally agree with the reviewer on this point. The recent Cryo-EM publications of RAF/14-3-3 complexes provided important insights to the RAF biology, therefore we have discussed these findings in the section about how 14-3-3 regulates the function of RAF (Line395-400).

It would be useful to include a couple of sentences on the N-terminal differences in RAF proteins (the region before RBD) and possible roles they may play in RAS/RAF biology.

This is a good suggestion. We have discussed how the N’-segment of RAF regulates the interaction of Ras/RAF (Fischer et al., 2007; Ding et al., 2010; Terrell et al., 2019) in the new version of our manuscript (line 262-264).

Reviewer 2 Report

No major corrections have to be made to the present review. Figures are illustrative and allow to understand the general mechanism involved in cancer initiation and progression.

Author Response

Reviewer #2

No major corrections have to be made to the present review. Figures are illustrative and allow to understand the general mechanism involved in cancer initiation and progression.

We thank reviewer #2 for his/her positive comments on our manuscript.
